# Control of Silver Coating on Raman Label Incorporated Gold Nanoparticles Assembled Silica Nanoparticles

**DOI:** 10.3390/ijms20061258

**Published:** 2019-03-13

**Authors:** Xuan-Hung Pham, Eunil Hahm, Eunji Kang, Byung Sung Son, Yuna Ha, Hyung-Mo Kim, Dae Hong Jeong, Bong-Hyun Jun

**Affiliations:** 1Department of Bioscience and Biotechnology, Konkuk University, Seoul 143-701, Korea; phamricky@gmail.com (X.-H.P.); greenice@konkuk.ac.kr (E.H.); ejkang@konkuk.ac.kr (E.K.); imsonbs@konkuk.ac.kr (B.S.S.); wes0510@konkuk.ac.kr (Y.H.); hmkim0109@konkuk.ac.kr (H.-M.K.); 2Department of Chemistry Education and Center for Educational Research, Seoul National University, Seoul 151-742, Korea; jeongdh@snu.ac.kr

**Keywords:** silver shell, silica template, Au–Ag alloy, nanogaps, SERS detection

## Abstract

Signal reproducibility in surface-enhanced Raman scattering (SERS) remains a challenge, limiting the scope of the quantitative applications of SERS. This drawback in quantitative SERS sensing can be overcome by incorporating internal standard chemicals between the core and shell structures of metal nanoparticles (NPs). Herein, we prepared a SERS-active core Raman labeling compound (RLC) shell material, based on Au–Ag NPs and assembled silica NPs (SiO_2_@Au@RLC@Ag NPs). Three types of RLCs were used as candidates for internal standards, including 4-mercaptobenzoic acid (4-MBA), 4-aminothiophenol (4-ATP) and 4-methylbenzenethiol (4-MBT), and their effects on the deposition of a silver shell were investigated. The formation of the Ag shell was strongly dependent on the concentration of the silver ion. The negative charge of SiO_2_@Au@RLCs facilitated the formation of an Ag shell. In various pH solutions, the size of the Ag NPs was larger at a low pH and smaller at a higher pH, due to a decrease in the reduction rate. The results provide a deeper understanding of features in silver deposition, to guide further research and development of a strong and reliable SERS probe based on SiO_2_@Au@RLC@Ag NPs.

## 1. Introduction

Surface-enhanced Raman scattering (SERS) has been widely used for various applications due to its excellent ultrasensitive molecular fingerprinting, and its non-destructive and photostable properties [1,2,3,4,5]. Much effort has been focused on the use of different nanoparticles (NPs) as a substrate for SERS detection, such as silver NPs [6,7], gold NPs [8,9,10,11], and metal-embedded graphene oxide [12,13]. Although these nanostructures can enhance the SERS signal, difficulty in controlling the density of hot spots on the surface of a SERS substrate makes them unsuitable for accurate quantitative SERS assays [14].

Internal standards have been used to correct variations in SERS intensity in quantitative SERS assays [14,15,16,17]. Internal standard-based quantitative SERS methods can be classified into three categories [14]: (i) internal standard addition detection mode [18,19]; (ii) internal standard tagging detection mode [20,21,22]; (iii) and ratiometric SERS indicator-based detection mode [14,23]. However, the concurrent presence of target molecules and internal standard compounds on the surface of a SERS-enhancing substrate can lead to the issue of competitive adsorption between the internal standard and the target analytes in both the addition and tagging detection modes. On the other hand, the ratiometric SERS indicator-based detection mode may avoid competition between the internal standard and target molecules, as the target molecules cannot adsorb onto the surface of the SERS substrate. However, difficulties in finding or synthesizing an appropriate SERS probe for a specified target have been a limiting factor in the general application of the ratiometric SERS indicator-based detection mode [14].

Core-shell nanomaterials have attracted attention and have been employed for various applications, such as solar cells [24,25,26], photocatalysis [27,28,29,30], sensors [31,32], biomedical diagnosis [33,34,35], and imaging [36,37]. This is due to their outstanding features [38], including versatility [39], economy [40], tunability [41,42], stability, dispersibility [43], biocompatibility [44], and controllability [45]. Since their localized surface plasmon resonance (LSPR) can become tunable by controlling the bimetallic component or structure, core-shell nanomaterials have been extensively used as a substrate to enhance Raman signals of probe molecules with exquisite sensitivity. The dynamic exchange between the target molecules and internal standard is bypassed, as the internal standard is embedded between the core and shell layers. However, the unstable sol form of “core-shell” substrates can cause faster agglomeration than solid substrates [46,47]. To overcome this problem, SERS-active core-Raman labeling chemical (RLC)-shell NPs (CRLCS NPs) have been used in SERS application, especially to avoid the competitive adsorption between the internal standard and target molecules, by embedding the internal standard in core-shell NPs as enhancing substrates [15,17,48,49]. Although the presence of RLC between the Au core and the Ag shell enables a strong and reliable SERS probe, to our knowledge the effect of RLC property on the growth of an Ag shell—which can be a critical factor in fabricating the homogeneous structure of core-shell materials—has not been investigated.

Recently, our group reported Au–Raman Labeling Chemical–Ag NP assembled silica NPs (SiO_2_@Au@RLC@Ag NPs) as strong and reliable SERS probes with an internal standard. SiO_2_@Au@RLC@Ag NPs were synthesized using an Au seed-mediated Ag growth method on the surface of a silica template, followed by incorporating RLC on their surfaces [50,51,52]. Herein, we investigated the effect of experimental conditions and RLC properties on the growth of an Ag shell on the surface of SiO_2_@Au. Three kinds of RLCs with a positive charge (4-aminothiphenol: 4-ATP), a negative charge (4-mercaptobenzoic acid: 4-MBA), and a neutral charge (4-methylbenzenthiol: 4-MBT) were used to investigate the effect of the charge properties of RLC on the growth of the Au shell. In addition, the influence of pH on the formation of the Ag shell was investigated.

## 2. Results and Discussion

To prepare SiO_2_@Au@RLC@Ag NPs, silica NPs (ca. 150 nm in diameter) were synthesized using the Stöber method [53] and used as a template for embedding the Au NPs. The surface of silica NPs was first functionalized with amine groups by (3-Aminopropyl) triethoxysilane (APTS) to prepare the aminated silica NPs, as shown in Figure 1. Simultaneously, colloidal Au NPs (7 nm) were prepared by NaBH_4_, according to the method reported by Martin et al., although with slight modifications [54,55]. Then, the Au NPs were incubated with the aminated silica NPs by gentle shaking to prepare an Au NPs embedded SiO_2_ (SiO_2_@Au NPs), since an amine functional group plays a crucial role in attaching the Au NPs through strong electrostatic attraction. Subsequently, three types of RLC with a positive charge (4-aminothiphenol: 4-ATP), a negative charge (4-mercaptobenzoic acid: 4-MBA) and a neutral charge (4-methylbenzenthiol: 4-MBT) were introduced on the surface of SiO_2_@Au NPs through the strong affinity between thiol groups and Au, to investigate the effect of charge properties of RLCs on the growth of the Au shell. Finally, the Ag shell was deposited on the SiO_2_@Au@RLC, to enhance the Raman signal of RLCs by reducing a silver precursor (AgNO_3_) in the presence of ascorbic acid and polyvinyl pyrrolidine (PVP) as a stabilizer and structure-directing agent under mild reducing conditions [51]. In addition, the presence of the Ag shell can prevent the leakage of RLC from the Au surface, and also provide a better chance of generating numerous hot spots on the silica surface to detect target molecules.

As expected, the Au NPs exhibited a typical UV peak at ~520 nm, as shown in Appendix A. After the Au NPs were coated on the surface of SiO_2_, the maximum peak of SiO_2_@Au was red-shifted to 530 nm. The zeta potential was used to confirm the result, and the SiO_2_ NPs had a zeta potential value of −44.6 ± 0.1 mV. When the surface of the SiO_2_ NP was incubated with APTS, the zeta potential value of SiO_2_@NH_2_ was increased to −27.7 ± 0.6 mV, due to the positive property of NH_2_ groups. Throughout the entire NH_2_ groups, the Au NPs were immobilized on the surface of SiO_2_@NH_2_ due to electrostatic attraction. Since the surface of the Au NPs was stabilized by BH_4_^-^, the zeta potential of SiO_2_@Au was decreased to −55.4 ± 6.1 mV (Appendix A).

### 2.1. Preparation of SiO_2_@Au@RLC@Ag

Three types of SiO_2_@Au@RLC@Ag nanomaterials with three different RLCs were successfully prepared in our study. The RLCs included 4-aminothiophenol (4-ATP) with a positive -NH_3_^+^ group; 4-MBA with a negative -COO^−^ group; and 4-methylbenzenethiol (4-MBT) with a neutral -CH_3_ group. The presence of -SH groups on their structures ensured that the RLCs bound to the surface of SiO_2_@Au, and exhibited their functional groups of -NH_3_^+^, -COO^−^, or -CH_3_ in the solution. As can be seen in Figure 2a, the structure of SiO_2_@Au@RLC@Ag was confirmed by the TEM analysis to show that the Ag shell was well coated on the surface of all RLCs-modified SiO_2_@Au.

The UV-Vis spectra of SiO_2_@Au@RLC@Ag were consistent with the TEM images (Figure 2b). In general, all solutions of SiO_2_@Au@RLC@Ag NPs showed a broad band from 320 to 800 nm, indicating the generation of bumpy structures on the Ag shell and the creation of hot-spot structures on the surface of SiO_2_@Au@RLC@Ag NPs [56]. At 300 µm AgNO_3_, a typical peak of SiO_2_@Au@RLCs was around 450 nm, due to the increase in the particle size of Au@RLC@Ag. However, the differences in the size of Au@Ag alloys and the distance of the nanogap between these alloys greatly affected their plasmon properties in the range of 700–800 nm, producing a continuous spectrum of resonant multimode [50,52,56,57,58,59]. The zeta potential of SiO_2_@Au@RLCs was measured (Appendix A) to explain the formation of the Ag shell on the surface of SiO_2_@Au@RLCs. As mentioned previously, the zeta potential of SiO_2_@Au was −55.4 ± 6.1 mV. When RLCs were modified on the surface of SiO_2_@Au, the zeta potential of all structures increased significantly. RLCs possess the -SH groups, which have a stronger affinity to Au NPs than NH_2_ groups on the surface of SiO_2_. Thus, RLCs may absorb on the surface of Au NPs, and some of the Au-RLC complex can migrate from the surface of SiO_2_@Au NPs, leading the zeta potential of RLCs-modified SiO_2_@Au NPs to be less negative. Yet, since the difference exists in functional groups of RLCs, SiO_2_@Au@RLC still possess a difference in surface charge of −35.2 ± 0.5 mV (4-ATP), −33.4 ± 1.3 mV (4-MBT) and −44.4 ± 6.9 mV (4-MBA), respectively. Nevertheless, the presence of negative charges on the surface of SiO_2_@Au@RLC facilitated the attraction of Ag^+^ ions to their surface and reduced them to Ag NPs.

Raman signals of three SiO_2_@Au@RLC@Ag nanomaterials were also measured (Figure 2c). The Raman intensity of SiO_2_@Au@4-MBA@Ag at 1075 cm^−1^ was the strongest compared to that of SiO_2_@Au@4-ATP@Ag and SiO_2_@Au@4-MBT@Ag. Raman signals of SiO_2_@Au@4-ATP@Ag and SiO_2_@Au@4-MBT@Ag were equal to those of the 68.3% and 7.9% of SiO_2_@Au@4-ATP@Ag, respectively.

### 2.2. Effect of Silver Ion Concentration on Ag Shell Coating on SiO_2_@Au@RLCs

To examine the effect of silver ion concentration on a silver shell coating of SiO_2_@Au@RLC, 4-MBA, 4-ATP, and 4-MBT were first introduced on the surface SiO_2_@Au NPs. The Ag shell was then deposited onto SiO_2_@Au@RLCs by the reduction of AgNO_3_, using ascorbic acid. The TEM analysis was performed to confirm the structure of SiO_2_@Au@RLC@Ag, as shown in Appendix A. When the AgNO_3_ concentration was increased from 50 to 300 µM, the size of Au@RLC@Ag alloy NPs became greater. However, Ag NPs (ca. 50–100 nm) appeared separately at higher concentrations of AgNO_3_ (>300 µM). This is possibly due to the formation of extra Ag NPs, made by nucleation in the solution during the reduction of high the AgNO_3_ concentration.

UV-Vis spectroscopies of the solution of SiO_2_@Au@RLC@Ag nanomaterials were recorded (Figure 3). The absorbance band of the SiO_2_@Au@RLC@Ag prepared with 4-ATP, 4-MBA, and 4-MBT appeared at 430–450 nm at low concentrations of AgNO_3_ (50 µM). The bands extended from 430 nm to 1000 nm when the AgNO_3_ concentration was increased to 700 µM. At the same time, their absorbance intensities were increased with a higher AgNO_3_ concentration. The results indicated that the silver shell was well coated on the surface of SiO_2_@Au@RLC in deionized water. Indeed, the Raman intensities of the SiO_2_@Au@RLC@Ag prepared with 4-ATP, 4-MBA, and 4-MBT became greater with an increase in the thickness of the Ag shell when AgNO_3_ increased from 50 µM to 200 µM. The Raman intensity plateaued when AgNO_3_ increased up to 300 µM. To compare the exact effects of Ag coating on the Raman signal of SiO_2_@Au@RLC@Ag without considering the differences in the intrinsic Raman properties of RLCs, we calculated the slopes of SiO_2_@Au@RLC@Ag in the range of 50 to 200 µM. The slopes of the normalized Raman signal were 0.105, 0.156, and 0.012 unit/µM, which correspond to 4-ATP, 4-MBA, and 4-MBT, respectively. The results indicate that the Ag shell coating significantly affected the Raman signals of these three SiO_2_@Au@RLC@Ag.

### 2.3. Effect of pH Solution on the Ag Shell Coating of SiO_2_@Au@RLC@Ag NPs

To confirm the effect of both pH and RLCs characteristics on the Ag shell coating of SiO_2_@Au@RLCs, we adjusted the pH of the solution during the reduction of Ag^+^. The coating of the Ag shell on the surface of SiO_2_@Au@RLCs was strongly dependent on the pH of the solution (Figure 4, Figure 5 and Figure 6). At a high pH, smaller sized silver nanoparticles were obtained, compared to those obtained at a low pH, due to the low reduction rate of AgNO_3_ precursors [60]. The coating of the Ag shell on the surface of SiO_2_@Au@4-MBT was rapid and worked well at a pH of 5.0, but became sluggish and difficult in acidic or basic pH values (Figure 4a and Appendix A). The Raman signals of SiO_2_@Au@4-MBT@Ag nanomaterials were measured (Figure 4b,c). The Raman signals of SiO_2_@Au@4-MBT@Ag were too weak and unclear because of small Au@4-MBT@Ag alloys with thin Ag shells. This result was consistent with the TEM images we observed in Figure 4a.

When 4-ATP was used as an RLC, the size of SiO_2_@Au@4-ATP@Ag became smaller when the pH was increased from 4.0 to 9.0 (Figure 5 and Appendix A). The coating of the Ag shell on the surface of SiO_2_@Au@4-ATP was rapid and worked well from an acidic to a basic pH solution. As a result, the Raman signals of SiO_2_@Au@4-ATP@Ag were observed clearly (Figure 5b,c). According to previous reports, pK_a_ values of 4-ATP on a gold surface range from 5.3 to 5.9 [61,62]. At a low pH (pH < 5), NH_2_ groups of 4-ATP on the surface of Au NPs exist in a protonated form (NH_3_^+^), and have a stronger affinity with Ag NPs generated in a bulk solution during the reduction of AgNO_3_ than with those generated during the deposition of the Ag shell on the surface of the SiO_2_@Au@4-ATP [63]. This may lead to the formation of large Ag NPs on the surface of SiO_2_@Au@4-ATP, as can be seen in TEM images (Appendix A), but did not significantly increase the Raman signal of 4-ATP (Figure 5). At a high pH (pH > 6), the deposition of the Ag shell on SiO_2_@Au@4-ATP dominated more, leading to a greater intensity of Raman signal in 4-ATP (Figure 5a).

Similarly, when 4-MBA was used as an RLC, the size of SiO_2_@Au@4-MBA@Ag became smaller when the pH was increased from 4.0 to 9.0 (Figure 6 and Appendix A). The coating of the Ag shell on the surface of SiO_2_@Au@4-MBA was also well obtained from an acidic to a basic pH solution. The carboxyl groups of 4-MBA existed in a protonated form (-COOH) at a low pH, lower than their pK_a_ (pK_a_ ≈ 5) [64,65,66]. The presence of -COOH inhibited the coating of the Ag shell on the surface of the SiO_2_@Au@4-MBA (Figure 6) and caused a low signal in 4-MBA (Figure 6). Similarly, the deprotonated form of the carboxylate groups (-COO^−^) became dominated on the surface of the SiO_2_@Au@4-MBA when the pH of the solution was raised and reached a value higher than the pK_a_ value of 4-MBA. They also led to an increase of the Raman signal of 4-MBA in the pH range of 5.0 to 6.0. It is known that, as the pH of solution increases continuously, silver oxide or silver chloride is formed [67], which can inhibit the coating of the Ag shell on the surface of SiO_2_@Au@4-MBA (Appendix A), with an obvious decrease in the Raman signal of 4-MBA from a pH of 7.0 to 9.0.

## 3. Experiment

### 3.1. Materials

Tetraethylorthosilicate (TEOS), 3-aminopropyltriethoxysilane (APTS), silver nitrate (AgNO_3_), chloroauric acid (HAuCl_4_), 4-mercaptobenzoic acid (4-MBA), ascorbic acid (AA), polyvinylpyrrolidone (PVP), sodium borohydride (NaBH_4_), and thiram were purchased from Sigma-Aldrich (St. Louis, MO, USA) and used without further purification. Ethyl alcohol (EtOH) and aqueous ammonium hydroxide (NH_4_OH, 27%) were purchased from Daejung (Siheung, Korea).

### 3.2. Preparation of SiO_2_@Au NP Templates

Silica NPs (~150 nm) were prepared using the Stöber method [53]. The silica NPs (50 mg mL^−1^, 4 mL) were dispersed in 4 mL of absolute EtOH, and 250 μL of APTS and 40 μL of NH_4_OH were added to the colloidal solution to aminate the silica NPs. The mixture was stirred vigorously for 6 h at 25 °C, followed by stirring for 1 h at 70 °C. The aminated silica NPs were obtained after centrifugation at 8500 rpm for 15 min, and then washed several times with EtOH to remove excess reagent.

The colloidal Au NPs were prepared by reducing HAuCl_4_, using NaBH_4_ as a reducing agent. The reduction of HAuCl_4_ created small Au NPs (~7 nm) with a net negative surface charge. In order to embed Au NPs into the silica NP surface, the Au NPs (1 mM, 10 mL) and aminated SiO_2_ solution (1 mg·mL^−1^, 1 mL) were mixed and sonicated for 30 min and incubated in a shaker overnight [50]. Then, Au NP-embedded silica NPs (SiO_2_@Au NPs) were obtained by centrifugation at 8500 rpm for 15 min, and washed several times with EtOH to remove unbound Au NPs. The SiO_2_@Au NPs were re-dispersed in absolute EtOH to obtain a SiO_2_@Au NP suspension of 1 mg·mL^−1^.

### 3.3. Incorporating RLC into SiO_2_@Au

RLC solution (1 mL, 10 mM in EtOH) was added to the SiO_2_@Au (1.0 mg), and the suspension was stirred vigorously for 2 h at 25°C. The colloids were centrifuged and washed several times with EtOH. The NPs were re-dispersed in 1.0 mL of absolute EtOH to obtain 1 mg·mL^−1^ SiO_2_@Au NPs modified with RLC (SiO_2_@Au@RLC).

### 3.4. Preparation of SiO_2_@Au@RLC@Ag NPs

Au-Ag core-shell NPs were prepared in an aqueous medium by the reduction and deposition of Ag with ascorbic acid onto the Au NPs in a polyvinylpyrrolidone (PVP) environment. Briefly, 0.2 mg of SiO_2_@Au@RLC was dispersed in 9.8 mL of water containing 10 mg PVP, and kept still for 30 min. Twenty microliters of 10 mM silver nitrate was added to the solution, followed by the addition of 20 µL of 10 mM ascorbic acid. This solution was incubated for 15 min to reduce the Ag^+^ ion to Ag. The reduction steps were repeated to obtain the desired AgNO_3_ concentration. SiO_2_@Au@4-MBA@Ag NPs were obtained by centrifugation of the solution at 8500 rpm for 15 min, and the NPs were washed several times with EtOH to remove excess reagent. SiO_2_@Au@4-MBA@Ag NPs were re-dispersed in 0.2 mL of absolute EtOH to obtain 1 mg·mL^−1^ SiO_2_@Au@4-MBA@Ag NP suspension.

### 3.5. SERS Measurement of the SiO2@Au@RLC@Ag NPs

SiO_2_@Au@RLC@Ag NPs were measured in a capillary tube, and SERS signals were measured using a confocal micro-Raman system (LabRam 300, JY-Horiba, Tokyo, Japan) equipped with an optical microscope (BX41, Olympus, Tokyo, Japan). The SERS signals were collected in a back-scattering geometry using a ×10 objective lens (0.90 NA, Olympus) and a spectrometer equipped with a thermoelectric cooled Charge-Coupled Device (CCD) detector. A 532 nm diode-pumped solid-state laser (CL532-100-S; Crystalaser, US) was used as a photo-excitation source, exerting 10 mW laser power at the sample. The strong Rayleigh scattered light was rejected using a long-pass filter. Selected sites were measured at random, and all SERS spectra were integrated for 5 s. The size of the laser beam spot was about 2 μm.

### 3.6. Transmission Electron Microscopy (TEM) Measurements

Our material was dispersed in EtOH to obtain a final concentration of 1 mg mL^−1^, and 10 µL of the dispersed solution was dropped onto a 400 mesh Cu grid (Pelco, Fresno, CA, USA) and dried in air. Field energy transmission electron microscopy (Libra 120, Carl Zeiss, Germany) was used to analyze our materials. The acceleration voltage was 120 kV.

## 4. Conclusions

In summary, we have prepared three types of SiO_2_@Au@RLC@Ag materials with three different RLCs, including 4-MBA, 4-ATP, and 4-MBT. The effect of RLCs on the deposition of the silver shell was also investigated. The formation of the Ag shell was strongly dependent on the negative charge of SiO_2_@Au@RLCs, the concentration of the silver ion, and the pH solution. In general, the size of Ag NPs was greater at a lower pH and became smaller at a higher pH due to the decrease in reduction rate. Especially, the pH of the solution played an important role in the formation of the Ag shell on the surface of SiO_2_@Au@RLCs, by affecting the local surface charge of the RLCs. For the neutral group of -CH_3_, the Ag shell was coated with difficulty on RLC-modified SiO_2_@Au, whereas the presence of the positive charge of -NH_3_^+^ on the surface of SiO_2_@Au facilitated the coating of the Ag shell, leading to a greater intensity of Raman signal in 4-ATP. The negative charge of -COO^−^ led to a well coated Ag shell, and increased the Raman signal of 4-MBA in the pH range of 5.0 to 6.0. However, it inhibited the coating of the Ag shell on the surface of SiO_2_@Au@4-MBA, with an obvious decrease in the Raman signal of 4-MBA from a pH of 7.0 to 9.0 due to the formation of silver oxide or silver chloride. This study provides a thorough understanding of silver deposition, to support further research and the development of strong and reliable SERS probes based on SiO_2_@Au@RLC@Ag NPs.

## Figures and Tables

**Figure 1 ijms-20-01258-f001:**
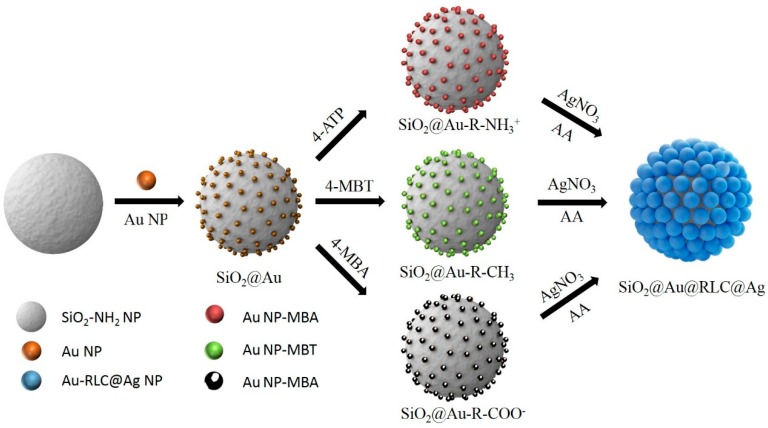
Illustration of a typical preparation of Au@Raman Labeling Compound@Ag embedded silica nanoparticles for a surface-enhanced Raman scattering (SERS) probe. Au NPs embedded silica nanoparticles were incubated with three different Raman labeling compounds, including 4-ATP, 4-MBA, and 4-MBT, and coated with an Ag shell by the reduction of silver nitrate in the presence of ascorbic acid and polyvinyl pyrrolidone.

**Figure 2 ijms-20-01258-f002:**
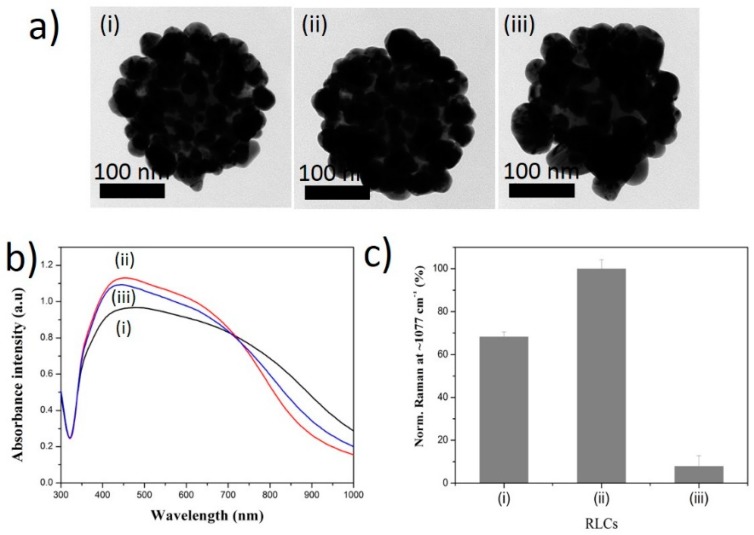
(**a**) Transmission electron microscopy (TEM) images, (**b**) UV-Vis absorption spectra of (i) SiO_2_@Au@4-ATP@Ag, (ii) SiO_2_@Au@4-MBA@Ag and (iii) SiO_2_@Au@4-MBT@Ag synthesized in water, and (**c**) their normalized Raman intensity at 1077 cm^−1^. All SiO_2_@Au was fixed at 200 µg. Concentration of Raman Labeling Chemical was 1 mM and that of AgNO_3_ was 300 µM.

**Figure 3 ijms-20-01258-f003:**
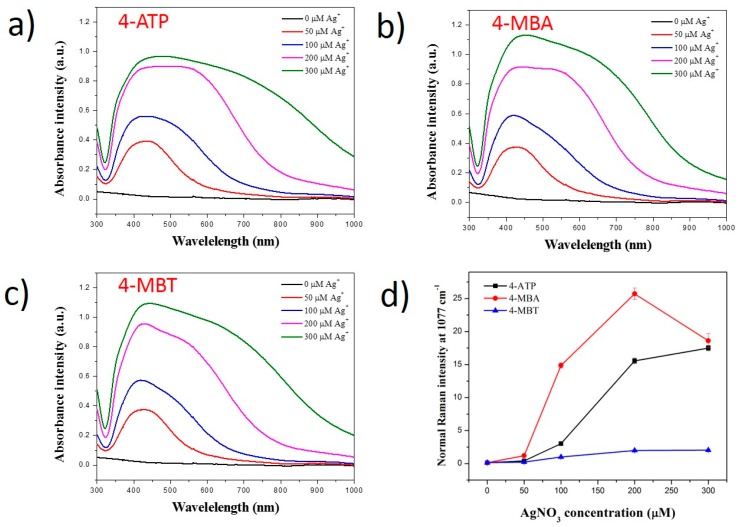
UV-Vis absorption spectra of (**a**) SiO_2_@Au@4-ATP@Ag, (**b**) SiO_2_@Au@4-MBA@Ag, (**c**) SiO_2_@Au@4-MBT@Ag nanoparticles, and (**d**) the normalized Raman spectra of the particles coated with different concentrations of AgNO_3_ in water. All SiO_2_@Au was fixed at 200 µg. Concentration of RLCs was 1 mM.

**Figure 4 ijms-20-01258-f004:**
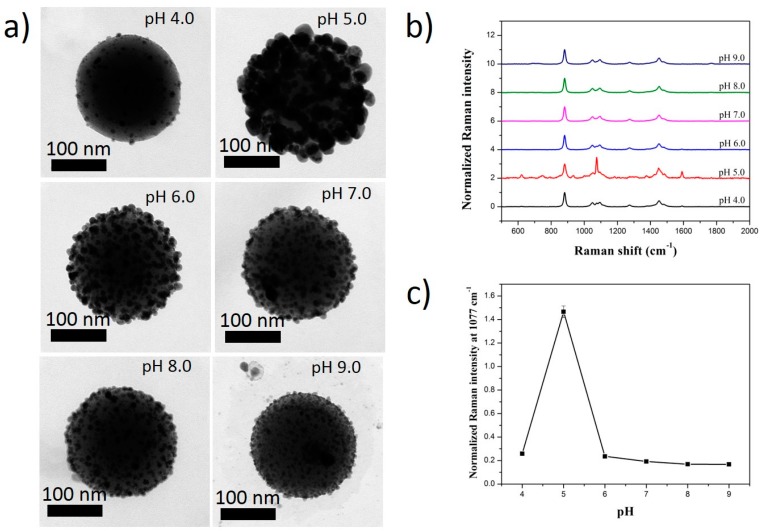
(**a**) TEM images and (**b**,**c**) Raman spectra of SiO_2_@Au@4-MBT@Ag synthesized at different pH solutions. All SiO_2_@Au was fixed at 200 µg. Concentration of RLCs was 1 mM and that of AgNO_3_ was 300 µM.

**Figure 5 ijms-20-01258-f005:**
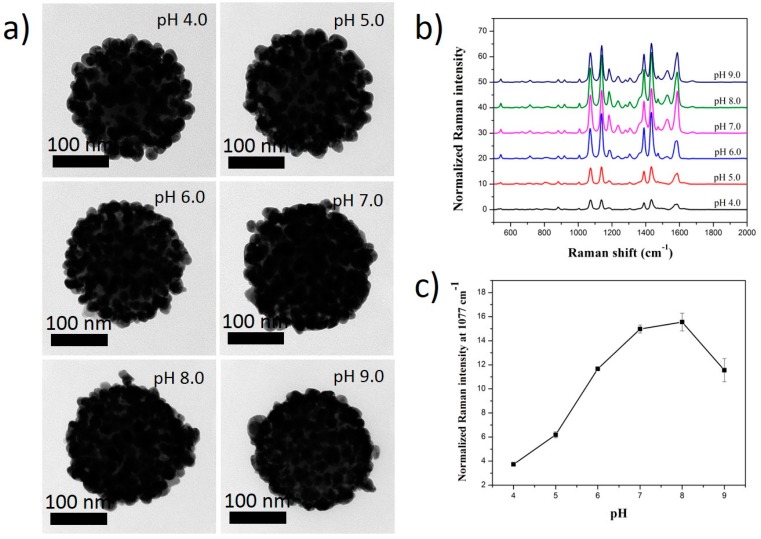
(**a**) TEM images and (**b**,**c**) Raman spectra of SiO_2_@Au@4-ATP@Ag synthesized at different pH solutions. All SiO_2_@Au was fixed at 200 µg. Concentration of RLCs was 1 mM and that of AgNO_3_ was 300 µM.

**Figure 6 ijms-20-01258-f006:**
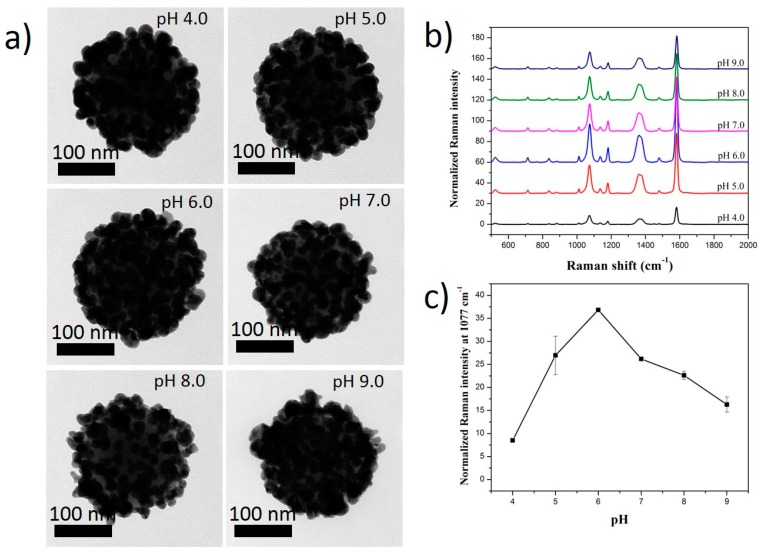
(**a**) TEM images and (**b**,**c**) Raman spectra of SiO_2_@Au@4-MBA@Ag synthesized at different pH solutions. All SiO_2_@Au was fixed at 200 µg. Concentration of RLCs was 1 mM and that of AgNO_3_ was 300 µM.

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
