# Peer review of "Control of Silver Coating on Raman Label Incorporated Gold Nanoparticles Assembled Silica Nanoparticles"

_ijms, 2019, doi:10.3390/ijms20061258_

Reviewer 1 Report

Dear authors

First of all, let me congratulate you for the work. Very interessting and of high level. Unfortunately I have some complains about the format you've used to structurate your paper and some comments to do:

- Page 2. Line 72 (until 88). You start with Results and discussion (in a very confusing mixture) when you've not still described Materials, Methodology. Instruments to be used. You give an enormous amount of information without a defined line of discurss. Everything (scientific concepts and experimental cases) is mixed. In my opinion, too much information to be "digested" for the reader. Indeed to me, that I'm familiar with all these techniques and procedures, has been difficult to follow. I've read several times before understanding, clearly, "what" and "how", to be capable to extract "why". It is a pitty for a so good work.

- Page 3. Scheme 1??? Which is the difference between Fig and Scheme. These fantastic squeme, in my opinion is just Fig. 1. Needs also, some more information and description attending to the physico-chemical phenomena that would be involved in the experiment. The squeme promises very interessting explanations, that are not included.

- Fig. 1. Part a) difficult to see differences between them. Is not possible to increase the size of TEM images?? Also, part b) will result more understandable if bigger. In my opinion the behaviour described is very important. Authors mentioned results in the range of 400 nm, but , what about the crossing phenomena of spectra in the range between 700 and 800 nm??? Curiously appears only in Fig. 1, but not on the rest of results (Fig. 2). Please, consider to try to explain also these range of wavelengths.

- Page 4. Line 120. "leakege of Au NPs from the surface of SiO2Au ionic exchange". It is the phrase that authors use for to explain modifications on zeta potential. I think that these changes need to be explained better and extensively. Means that, Au Complexed migrates to outside? Is this charge compensed by NPs? Do authors think that there can be some diffusion phenomena?

- Page 6. Fig 3: To help readers to understand the sensibility to pH, I recommend to separate each of the pictures in three different Fig's and explained separately. From results, apparently, the mechanisms proposed in "Squeme 1", fit perfectly with experiments. Why not to comment that? If the explanation of "Squeme 1" was more phenomenologic, then results can be understood better.

- Page 7. Line 195. All this paragraph should be moved to paragraph 2.

- Page 8: In my opinion, with a so good results obtained, conclusions are very poor. I miss in the explanations a more phemomenological approach.

With so high skills work showed by authors, I think that it is possible to get a more fruitful explanations for readers.

Sorry for the inconveniences caused, my aim is only to improve your magnificient work presented here

Author Response

Dear reviewers

We appreciate the comments from the reviewers who spent invaluable time and effort. We have incorporated additional modifications based on the reviewer’s thoughtful comments, which have helped us to improve the manuscript. The detailed responses to the reviewer’s comments are provided at the following.

Point 1: Page 2. Line 72 (until 88). You start with Results and discussion (in a very confusing mixture) when you've not still described Materials, Methodology, Instruments to be used. Let’s start with Materials, Methodology, and Instruments. You give an enormous amount of information without a defined line of discuss. Everything (scientific concepts and experimental cases) is mixed. In my opinion, too much information to be "digested" for the reader. Indeed to me, that I'm familiar with all these techniques and procedures, has been difficult to follow. I've read several times before understanding, clearly, "what" and "how", to be capable to extract "why". It is a pitty for a so good work.

Response 1: Thank you for your suggestion. We moved Materials and Method to section 2 in the revised manuscript. Also, we rewrote the first paragraph of section 3 Results and Discussion to make our story clear as your suggestion. (line 125 -140)

“To prepare SiO2@Au@RLC@Ag NPs, silica NPs (ca. 150 nm in diameter) were synthesized by the Stöber method[53] and used as a template for embedding the AuNPs. The surface of silica NPs was first functionalized with amine groups by APTS to prepare the aminated silica NPs as shown in Figure 1. Simultaneously, colloidal AuNPs (7 nm) were prepared by NaBH4, according to the method reported by Martin et al. with slight modifications.[54, 55] Then, the AuNPs were incubated with the aminated silica NPs by gentle shaking to prepare an AuNPs embedded SiO2 (SiO2@Au NPs). Since an amine functional group plays a crucial role in attaching the AuNPs through strong electrostatic attraction. Subsequently, three types of RLC with a positive charge (4-aminothiphenol: 4-ATP), a negative charge (4-mercaptobenzoic acid: 4-MBA) and a neutral charge (4-methylbenzenthiol: 4-MBT) were introduced on the surface of SiO2@Au NPs through strong affinity between thiol groups and Au to investigate the effect of charge properties of RLCs on the growth of the Au shell. Final, the Ag shell was deposited on the SiO2@Au@RLC to enhance the Raman signal of RLCs by reducing a silver precursor (AgNO3) in the presence of ascorbic acid and PVP as a stabilizer and structure-directing agent under mild reducing conditions.[51] In addition, the presence of the Ag shell can prevent the leakage of RLC from the Au surface and also provide a better chance to generate numerous hot spots on the silica surface to detect target molecules.”

 Point 2: Page 3. Scheme 1??? Which is the difference between Fig and Scheme. These fantastic squeme, in my opinion is just Fig. 1. Needs also, some more information and description attending to the physico-chemical phenomena that would be involved in the experiment. The squeme promises very interesting explanations, that are not included.

Response 2: Thank you for your suggestion. We changed the scheme 1 to figure 1, and added more information and description in Figure 1 (line 142-145)

Figure 1. Illustration of typical preparation of Au@Raman Labeling Compound@Ag embedded silica nanoparticles for SERS probe. Au NPs embedded silica nanoparticles were incubated with three different Raman Labeling Compound including 4-ATP, 4-MBA and 4-MBT and coated with Ag shell by reduction of silver nitrate in the presence of ascorbic acid and polyvinyl pyrrolidone.

Point 3: Fig. 1. Part a) difficult to see differences between them. Is not possible to increase the size of TEM images?? Also, part b) will result more understandable if bigger. In my opinion the behaviour described is very important. Authors mentioned results in the range of 400 nm, but , what about the crossing phenomena of spectra in the range between 700 and 800 nm???

Response 3: Thank you for your comment.

We agreed that it is difficult to recognize the difference of three TEM images of SiO2@Au@RLC@Ag NPs in part a. Therefore, we measured both UV-Vis spectra and Raman intensities of these SERS probes to confirm their difference characteristic in part b and c.

In part b, the presence of Au@RLC@Ag alloys on the surface of RLC-modified SiO2@Au were confirm by the UV-Vis spectra in the range of 450 nm. However, the differences in the size of Au@Ag alloys and the distance of nanogap between these alloys greatly affected their plasmon properties in the range of 700-800 nm, which assigned to a continuous spectrum of resonant multimode reported by our groups Bastús et al., and our previous report.

N.G. Bastús, F. Merkoçi, J. Piella, V. Puntes, Chem Mater, 26 (2014) 2836-46.

X.-H. Pham, M. Lee, S. Shim, S. Jeong, H.-M. Kim, E. Hahm, S.H. Lee, Y.-S. Lee, D.H. Jeong, B.-H. Jun, RSC Adv., 7 (2017) 7015-7021.

 Point 4: Authors mentioned results in the range of 400 nm, but , what about the crossing phenomena of spectra in the range between 700 and 800 nm??? Curiously appears only in Fig. 1, but not on the rest of results (Fig. 2). Please, consider to try to explain also these range of wavelengths.

Response 4: In Fig. 1 we plotted three spectra of SiO2@Au@4-ATP@Ag, SiO2@Au@4-MBA@Ag and SiO2@Au@4-MBT@Ag NPs together. So, we can see the crossing phenomena as you mentioned. However, the UV-Vis spectra of SiO2@Au@4-ATP@Ag, SiO2@Au@4-MBA@Ag and SiO2@Au@4-MBT@Ag NPs were plotted separately coated with different concentrations of AgNO3 in water AgNO3 concentration in Figure 2a, 2b and 2c. As can be seen in all figures, the UV-Vis spectra of the SiO2@Au@RLC@Ag appeared at 430–450 nm at low concentration of AgNO3. The bands extended from 430 nm to 1000 nm when AgNO3 concentration increased.

To make our point clearly, we modified the sentence in revised manuscript (line 165 -169)

“The UV-Vis spectra of SiO2@Au@RLC@Ag were consistent with the TEM images (Figure 2b). In general, all solution of SiO2@Au@RLC@Ag NPs showed a broad band from 320 to 800 nm, indicating generation of bumpy structures on the Ag shell and creation of hot-spot structures on the surface of SiO2@Au@RLC@Ag NPs [56]. At 300 µm AgNO3, a typical peak of SiO2@Au@RLCs was around 450 nm due to the increase in the particle size of Au@RLC@Ag. However, the differences in the size of Au@Ag alloys and the distance of nanogap between these alloys greatly affected their plasmon properties in the range of 700-800 nm, producing a continuous spectrum of resonant multimode was also identified.[50, 52, 56-59]

 Point 5: Page 4. Line 120. "leakage of Au NPs from the surface of SiO2@Au by ionic exchange". It is the phrase that authors use for to explain modifications on zeta potential. I think that these changes need to be explained better and extensively. Means that, Au Complexed migrates to outside? Is this charge compensed by NPs? Do authors think that there can be some diffusion phenomena?

Response 5: Thank you for your comment. We agreed that it is not simply that AuNPs leaked out from the surface of SiO2 NPs will alter the zeta potential of SiO2@Au@RLCs.

As we mentioned in our manuscript, the charge surfaces of both SiO2 and AuNPs are negative. Because AuNPs attached to SiO2 NPs through electrostatic attraction of the NH2 groups on the surface of SiO2. All RLCs possesses the -SH groups, which have strong affinity to AuNPs. Thus, we guessed that RLCs can absorbed on the surface of AuNPs and some of Au-RLC complex can migrate to the bulk.

In addition, some possible diffusion phenomena in this process that we have not yet discovered.

Therefore, we modified our sentence in revised manuscript (line 156-161).

“When RLCs were modified on the surface of SiO2@Au, the zeta potential of all structures increased significantly. RLCs possesses the -SH groups, which have stronger affinity to AuNPs than NH2 groups on the surface of SiO2. Thus, RLCs may absorbed on the surface of AuNPs and some of Au-RLC complex can migrate from the surface of SiO2@Au NPs, leading the zeta potential of RLCs-modified SiO2@Au NPs to be less negative.

 Point 6: Page 6. Fig 3: To help readers to understand the sensibility to pH, I recommend to separate each of the pictures in three different Fig's and explained separately. From results, apparently, the mechanisms proposed in "Squeme 1", fit perfectly with experiments. Why not to comment that? If the explanation of "Squeme 1" was more phenomenologic, then results can be understood better.

Response 6: It is an excellent idea. Thank you for your suggestion.

We combine Figure 3 and Figure 4 in the old version to prepare Figure 4, 5 and 6 in the revised manuscript. Also, we rewrote some paragraphs to explain the results in these figures (line 219-228, line 233-243, line 248 -259).

To confirm the effect of both pH and RLCs characteristics on the Ag shell coating of SiO2@Au@RLCs, we adjusted the pH of solution during the reduction of Ag+. Coating of the Ag shell on the surface of SiO2@Au@RLCs was strongly dependent on the pH of solution (Figure 4 – Figure 6). At high pH, smaller sized silver nanoparticles were obtained, compared to those obtained at low pH, due to the low reduction rate of AgNO3 precursors [60]. The coating of the Ag shell on the surface of SiO2@Au@4-MBT was rapid and well at pH of 5.0 but became sluggish and difficult in acidic or basic pH values (Figure 4a and Figure S6). Raman signals of SiO2@Au@4-MBT@Ag nanomaterials were measured (Figure 4b-c). The Raman signals of SiO2@Au@4-MBT@Ag was too weak and unclear because of small Au@4-MBT@Ag alloys with thin Ag shell. This result was consistent with the TEM images we observed in Figure 4a.

When 4-ATP were used as an RLC, the size of SiO2@Au@4-ATP@Ag became smaller when pH was increased from 4.0 to 9.0 (Figure 5 and Figure S7). The coating of the Ag shell on the surface of SiO2@Au@4-ATP was rapid and well from acidic to basic pH solution. As a result, the Raman signals of SiO2@Au@4-ATP@Ag were observed clearly (Figure 5b-c). According to previous reports, pKa values of 4-ATP on a gold surface range from 5.3 to 5.9 [61, 62]. At low pH (pH < 5), NH2 groups of 4-ATP on the surface of Au NPs exist in a protonated form (NH3+) and have a stronger affinity with Ag NPs generated in bulk solution during reduction of AgNO3 than during deposition of the Ag shell on the surface of the SiO2@Au@4-ATP.[63] This may lead to the formation of large-sized Ag NPs on the surface of SiO2@Au@4-ATP, as can be seen TEM images (Figure S7), but did not significantly increase the Raman signal of 4-ATP (Figure 5). At high pH (pH > 6), the deposition of the Ag shell on SiO2@Au@4-ATP dominated more, leading to greater intensity of Raman signal in 4-ATP (Figure 5a).

Similarly, when 4-MBA were used as an RLC, the size of SiO2@Au@4-MBA@Ag became smaller when pH was increased from 4.0 to 9.0 (Figure 6 and Figure S8). The coating of the Ag shell on the surface of SiO2@Au@4-ATP was also well from acidic to basic pH solution. the carboxyl groups of 4-MBA existed in a protonated form (-COOH) at low pH, lower than their pKa (pKa » 5) [64-66]. The presence of COOH inhibited the coating of the Ag shell on the surface of the SiO2@Au@4-MBA (Figure 4b) and caused a low signal in 4-MBA (Figure 4). Similarly, the deprotonated form of carboxylate groups (-COO-) became dominated on the surface of the SiO2@Au@4-MBA when pH of solution was raised and reached a value higher than the pKa value of 4-MBA. They also led to an increase of Raman signal of 4-MBA in the range of pH between 5.0 to 6.0. It is known that as the pH of solution increases continuously, silver oxide or silver chloride is formed [67], which can inhibit the coating of the Ag shell on the surface of SiO2@Au@4-MBA (Figure S8) with an obvious decrease of Raman signal of 4-MBA from pH of 7.0 to 9.0.

 Point 7: Page 7. Line 195. All this paragraph should be moved to paragraph 2.

Response 7: Thank you for your suggestion. We moved this paragraph to section 2 Materials and Method in the revised manuscript.

 Point 8: Page 8: In my opinion, with a so good results obtained, conclusions are very poor. I miss in the explanations a more phemomenological approach.

Response 8: Thank you for your comment. We rewrote our conclusion to explain our story more phemomenological approach (line 266-280).

In summary, we have prepared three types of SiO2@Au@RLC@Ag materials with three different RLCs including 4-MBA, 4-ATP and 4-MBT. The effect of RLCs on the deposition of the silver shell was also investigated. The formation of the Ag shell was strongly dependent on the negative charge of SiO2@Au@RLCs, the concentration of silver ion and pH solution. In general, the size of Ag NPs was greater at lower pH and became smaller at higher pH due to the decrease in reduction rate. Especially, pH of solution played an important role in the formation of the Ag shell on the surface of SiO2@Au@RLCs by affecting on the local surface charge of RLCs. For neutral group of -CH3, Ag shell was difficultly coated on RLC-modified SiO2@Au. Whereas, the presence of positive charge of -NH3+ on the surface of SiO2@Au facilitated the coating of Ag shell leading to greater intensity of Raman signal in 4-ATP. The negative charge of -COO- led a well coated of Ag shell and increased Raman signal of 4-MBA in the range of pH between 5.0 to 6.0 but inhibit the coating of the Ag shell on the surface of SiO2@Au@4-MBA with an obvious decrease of Raman signal of 4-MBA from pH of 7.0 to 9.0 because the formation of silver oxide or silver chloride. This study provides a thorough understanding of silver deposition to support further research and development of strong and reliable SERS probes based on SiO2@Au@RLC@Ag NPs.

 Reviewer 2 Report

     The results are presented in this paper allow provide a deeper understanding of features in silver deposition which is imprtant to guide further research and development of a strong and reliable SERS probe based on SiO2/Au/RLC/AgNPs.

 The paper requires the following minor amendments:

 ·       in the whole manuscript, the acronym recording method should be changed: is Ag NPs or Au NPs should be AgNPs or AuNPs.

·       line 107: should be -CH3 as in the case of -SH.

·       lines 185-186: carboxyl group: is OOCH, should be -OOCH
                          which means; (pKa  5)

 Summarizing: the reviewed paper may be recommended for publication in International Journal of Molecular Science as a Full Paper after minor alterations.

Author Response

Dear reviewers

We appreciate the comments from the reviewers who spent invaluable time and effort. We have incorporated additional modifications based on the reviewer’s thoughtful comments, which have helped us to improve the manuscript. The detailed responses to the reviewer’s comments are provided at the following.

Point 1: in the whole manuscript, the acronym recording method should be changed: is Ag NPs or Au NPs should be AgNPs or AuNPs.

Response 1: Thank you for your comment, we corrected the acronym Ag NPs or Au NPs to be AgNPs and AuNPs as your suggestion.

 Point 2:  line 107: should be -CH3 as in the case of -SH.

Response 2: Thank you for your comment. MBT stands for 4-methylbenzenethiol, which possesses -SH group and -CH3 group. Thiol group of MBT connect with AuNPs and -CH3 group turn outward to solution.

Point 3: lines 185-186: carboxyl group: is OOCH, should be -OOCH

Response 3: Thank you for your comment.

We changed the phrase to “carboxyl groups of 4-MBA existed in a protonated form (-COOH)”

Point 4: which means; (pKa » 5)

 Response 4: According to reference 64, the pKa of free 4-MBA is 4.16. However, when individual 4-MBA is coated on the surface of AuNPs, this value is in the range of 5.18 -7.58, depending on the position of ligand.

Round  2

Reviewer 1 Report

Dear Authors

Congratulations, your work, now can be better understood. Thanks a lot for the efforts devoted to follow the advices of the first revission